# Live-dead assay on unlabeled cells using phase imaging with computational specificity

Chenfei Hu [1,2,8], Shenghua He[3,8], Young Jae Lee [2,4], Yuchen He [1,2], Edward M. Kong[2], Hua Li[5,6,7], Mark A. Anastasio[2,5 ✉] & Gabriel Popescu [1,2,5 ✉]

Existing approaches to evaluate cell viability involve cell staining with chemical reagents. However, the step of exogenous staining makes these methods undesirable for rapid, non-destructive, and long-term investigation. Here, we present an instantaneous viability assessment of unlabeled cells using phase imaging with computation specificity. This concept utilizes deep learning techniques to compute viability markers associated with the specimen measured by label-free quantitative phase imaging. Demonstrated on different live cell cultures, the proposed method reports approximately 95% accuracy in identifying live and dead cells. The evolution of the cell dry mass and nucleus area for the labeled and unlabeled populations reveal that the chemical reagents decrease viability. The nondestructive approach presented here may find a broad range of applications, from monitoring the production of biopharmaceuticals to assessing the effectiveness of cancer treatments.

[1] Electrical and Computer Engineering, University of Illinois at Urbana-Champaign, Urbana, IL, USA. [2] Beckman Institute for Advanced Science and Technology, University of Illinois at Urbana-Champaign, Urbana, IL, USA. [3] Department of Computer Science & Engineering, Washington University in St. Louis, St. Louis, MO, USA. [4] Neuroscience Program, University of Illinois at Urbana-Champaign, Urbana, IL, USA. [5] Department of Bioengineering, University of Illinois at Urbana-Champaign, Urbana, IL, USA. [6] Cancer Center at Illinois, Urbana, IL, USA. [7] Department of Radiation Oncology, Washington University in St. Louis, Saint Louis, MO, USA. [8] These authors contributed equally: Chenfei Hu, Shenghua He. ✉email: maa@illinois.edu; gpopescu@illinois.edu

Rapid and accurate estimation of the viability of biological cells is important for assessing the impact of drugs, physical or chemical stimulants, and other potential factors in cell function. The existing methods to evaluate cell viability commonly require mixing a population of cells with reagents to convert a substrate to a colored or fluorescent product[1]. For instance, using membrane integrity as an indicator, the live and dead cells can be separated by trypan blue exclusion assay, where only nonviable cells are stained and appear as a distinctive blue color under a microscope[2,3]. MTT and XTT assay estimate the viability of a cell population by measuring the optical absorbance caused by formazan concentration due to alteration in mitochondrial activity[4–6]. Starting in the 1970s, fluorescence imaging has developed as a more accurate, faster, and reliable method to determine cell viability[7–10]. Similar to the principle of trypan blue test, this method identifies individual nonviable cells by using fluorescent reagents only taken up by cells that lost their membrane permeability barrier. Unfortunately, the step of exogenous labeling generally requires some incubation time for optimal staining intensity, making all these methods difficult for quick evaluation. Importantly, the toxicity introduced by stains eventually kills the cells and, thus, prevents the long-term investigation.

Quantitative phase imaging (QPI) is a label-free modality that has gained significant interest due to its broad range of potential biomedical applications[11,12]. QPI measures the optical phase delay across the specimen as an intrinsic contrast mechanism, and thus, allows visualizing transparent specimen (i.e., cells and thin tissue slices) with nanometer scale sensitivity, which makes this modality particularly useful for nondestructive investigations of cell dynamics (i.e. growth, proliferation, and mass transport) in both 2D and 3D[13–18]. In addition, the optical phase delay is linearly related to the non-aqueous content in cells (referred to as dry mass), which directly yields biophysical properties of the sample of interest[19–22]. More recently, with the concomitant advances in deep learning, we have witnessed exciting avenues for label-free imaging. In 2018, Google presented "in silico labeling", a deep learning based approach that can predict fluorescent labels from transmitted-light (bright field and phase contrast) images of unlabeled samples[23]. Around the same time, researchers from the Allen Institute showed that individual subcellular structures such as DNA, cell membrane, and mitochondria can be obtained computationally from bright-field images[24]. Because a QPI map quantitatively encodes structure and biophysical information, it is possible to apply deep learning techniques to extract subcellular structures[25,26], perform signal reconstruction[27,28], correct image artifacts[29,30], convert QPI data into virtually stained or fluorescent images[31,32], and diagnose and classify various specimens[33,34].

In this article, we demonstrate that rapid viability assay can be conducted in a label-free manner using spatial light interference microscopy (SLIM)[35,36], a highly sensitive QPI method, and deep learning. We apply the concept of phase imaging with computational specificity (PICS) to digitally stain for the live and dead markers. Demonstrated on live adherent HeLa and CHO cell cultures, we predict the viability of individual cells measured with SLIM by using a joint EfficientNet[37] and transfer learning[38] strategy. Using the standard fluorescent viability imaging as ground truth, the trained neural network classifies the viable state of individual cell with 95% accuracy. Furthermore, by tracking the cell morphology over time, unstained HeLa cells show significantly higher viability compared to the cells stained with viability reagents. These findings suggest that the PICS method enables rapid, nondestructive, and unbiased cell viability assessment, potentially valuable to a broad range of biomedical problems, from drug testing to the production of biopharmaceuticals.

## Results

The procedure of image acquisition is summarized in Fig. 1. We employed spatial light interference microscopy (SLIM)[35] to measure the quantitative phase map of cells in vitro. The system is built by attaching a SLIM module (CellVista SLIM Pro, Phi Optics, Inc.) to the output port of an existing phase-contrast microscope (Fig. 1a). By modulating the optical phase delay between the incident and the scattered field, a quantitative phase map is retrieved from four intensity images via phase-shifting

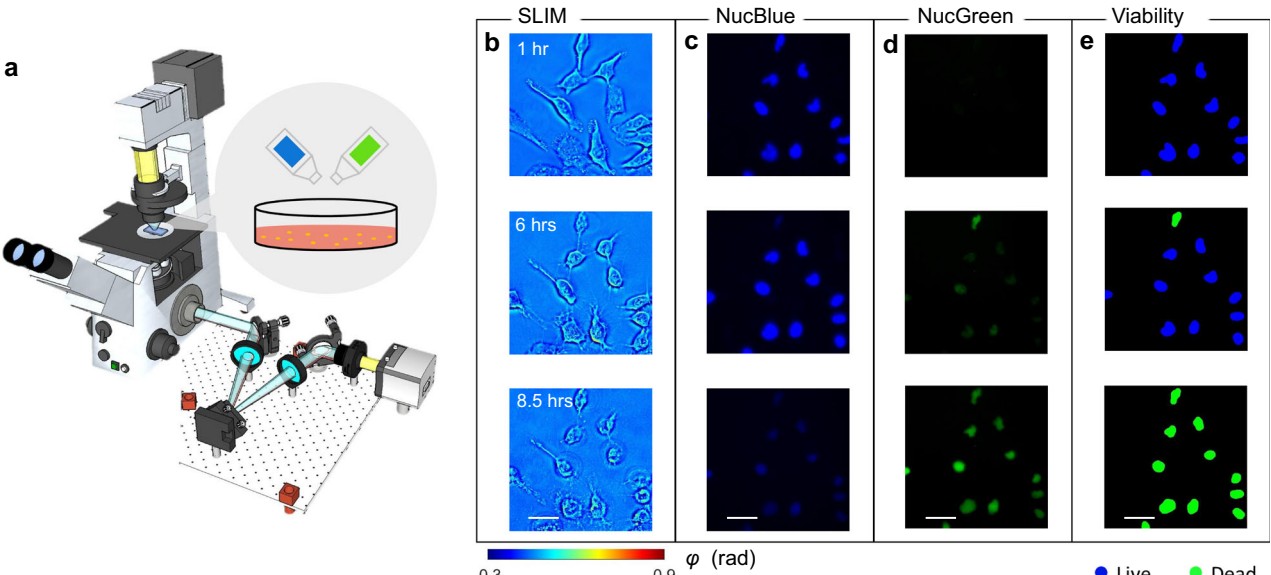

**Fig. 1 Schematic of the imaging system and representative results. a** CellVista SLIM Pro microscope (Phi Optics, Inc.) consists of an existing phase contrast microscope and an external module attached to the output port. By switching between transmission and reflection excitation, both SLIM and co-localized fluorescence images can be recorded via the same optical path. Before time-lapse imaging started, fluorescence viability reagents were mixed with the cell culture. **b** Representative SLIM measurements of HeLa cell at 1, 6, and 8.5 h after staining. The experiment is repeated 4 times. **c** NucBlue fluorescent signals of the live viability reagent. **d** NucGreen fluorescent signals of the dead viability reagents measured. **e** Viability states of the individual cells. Source data are provided as a Source Data file. Scale bars: 50 μm in space.

interferometry[39]. SLIM employs broadband LED as an illumination source and common-path imaging architecture, which yields sub-nanometer sensitivity to optical pathlength changes and high temporal stability[39,40]. By switching to epi-illumination, the optical path of SLIM is also used to record the fluorescent signals over the same field of view. Detailed information about the microscope configuration can be found in Methods.

To demonstrate the feasibility of the proposed method, we imaged and analyzed live cell cultures. Before imaging, 40 μL of each cell-viability-assay reagent (ReadyProbes Cells Viability Imaging Kit, Thermofisher) was added into 1 ml growth media, and the cells were then incubated for approximately 15 min to achieve optimal staining intensity. The viability-assay kit contains two fluorescently labeled reagents: NucBlue (the "live" reagent) combines with the nuclei of all cells and can be imaged with a DAPI fluorescent filter set, and NucGreen (the "dead" reagent) stains the nuclei of cells with compromised membrane integrity, which is imaged with a FITC filter set. In this assay, live cells produce a blue-fluorescent signal; dead cells emit both green and blue fluorescence; The procedure of cell culture preparation can be found in Methods.

After staining, the sample was transferred to the microscope stage, and measured by SLIM and epi-fluorescence microscopy. In order to generate a heterogeneous cell distribution that shifts from predominantly alive to mostly dead cells, the imaging was performed under room conditions, such that the low-temperature and imbalanced pH level in the media would adversely injure the cells and eventually cause necrosis. Recording one measurement every 30 or 60 min, the entire imaging process lasted for approximately 10 h. We repeated this experiment four times to capture the variability among different batches. Figure 1b shows the SLIM images of HeLa cells measured at t = 1 h, 6, and 8.5 h, respectively, and the corresponding fluorescent measurements are shown in Fig. 1c, d. The results in Fig. 1 show that the adverse environmental condition continues injuring the cell, where blebbing and membrane disruption could be observed during cell death. Our QPI measurements agree with the results reported in previous literature[41]. On the other hand, these morphological alterations are correlated with the changes in fluorescence signals, where the intensity of NucGreen ("dead" fluorescent channel) continuously increases, as cells transit to dead states. By comparing the relative intensity between NucGreen and NucBlue signals, semantic segmentation maps are generated to label individual cell as either live or dead, as shown in Fig. 1e. The procedure of generating the semantic maps can be found in Supplemental Note 1. All collected image sequences were combined to form a dataset for PICS training and testing, where each sequence is a time-lapse recording of cells from live to dead states. Then we randomly split the sequences with a ratio of approximately 6:1:1, to obtain training, validation, and testing dataset, respectively. Instead of splitting by frame, we generated a training dataset by dividing image sequences to ensure fair generalization. In addition, we combined data across all measurements to take underrepresented cellular activities into account, which makes the purposed method generalizable.

**Deep neural network architecture, training, validation, and testing**. With fluorescence-based semantic maps as ground truth, a deep neural network was trained to assign "live", "dead", or background labels to pixels in the input SLIM images. We employed a U-Net based on EfficientNet (E-U-Net)[37], with its architecture shown in Fig. 2a. Compared to conventional U-Nets, the E-U-Net uses EfficientNet[37], a powerful network of relatively lower complexity, as the encoding part. This architecture allows for learning an efficient and accurate end-to-end segmentation

model, while avoiding training a very complex network. The network was trained using a transfer learning strategy[38] with a finite training set. At first, the EfficientNet of E-U-Net (the encoding part) was pre-trained for image classification on a publicly available dataset ImageNet[42]. The entire E-U-Net was then further fine-tuned for a semantic segmentation task by using labeled SLIM images from the training and validation set.

The network training was performed by updating the weights of parameters in the E-U-Net using an Adam optimizer[43] to minimize a loss function that is computed in the training set. More details about the EfficientNet module and loss function can be found in the Methods and Supplemental Note 2. The network was trained for 100 epochs. At the end of each epoch, the loss function related to the being-trained network was evaluated, and the weights that yielded the lowest loss on the validation set were selected for the E-U-Net model. Figure 2d shows training and validation loss vs. the number of epochs, using 899 and 199 labeled images as training and validation datasets. The Methods section and Fig. 2a–c present more details about the E-U-Net architecture and network training.

To demonstrate the performance of phase imaging with computational specificity (PICS) as a label-free live/dead assay, we applied the trained network to 200 SLIM images not used in training and validation. Figure 3a shows the three representative testing phase maps, whereas corresponding ground truth and PICS prediction are shown in Fig. 3b, c, respectively. This direct comparison indicates that PICS successfully classifies the cell states. We found that, most often, the incorrect predictions were caused by cells located at the boundary of FOV, where only a portion of their cell bodies was measured by SLIM. In addition, PICS may fail when cells become detached from the well plates. In this situation, the suspended cells appear out of focus, which gives rise to inaccurate prediction. As reported in previous publications, the conventional deep learning evaluation metrics focus on assessing pixel-wise segmentation accuracy, which overlooks some biologically relevant instances[44]. Here, we adopted an object-based evaluation metric, which relies on comparing the dominant semantic label between the predicted cell nuclei and the ground truth for individual nuclei. The confusion matrix and the corresponding evaluation (e.g., precision, recall, and F1-score) are shown in Table 1. A comparison with standard pixel-wise evaluation and procedure of object-based evaluation are included in Supplemental Note 3. The entries of the confusion matrix are normalized with respect to the number of cells in each category. Using the average F1 score across all categories as an indicator of the overall performance, this PICS strategy reports a 96.7% confidence in distinguishing individual live and dead HeLa cells.

**PICS on CHO cells**. Chinese hamster ovary (CHO) cells are often used for recombinant protein production, and it received U.S. FDA approval for bio-therapeutic protein production. Here, we demonstrate that our label-free viability assay approach is applicable to other cell lines of interest in pharmaceutical applications. CHO cells were plated on a glass-bottom 6-well plate for optimal confluency. In addition to NucBlue/NucGreen staining, 1 μM of staurosporine (apoptotic inducing reagent) solution was added to the culture medium. This potent reagent permeates the cell membrane and disrupts protein kinase, cAMP, and leads to apoptosis in 4–6 h. The cells were then measured by SLIM and epi-fluorescence microscopy. The cells were maintained in regular incubation conditions (37 °C and 5% concentration of $CO_2$) throughout the experiment. In addition, we verified that the cells were not affected by necrosis and lytic cell death (see Supplemental Note 4). After image acquisition, E-U-Net (EfficientNet-B7) training was immediately followed. In the training process,

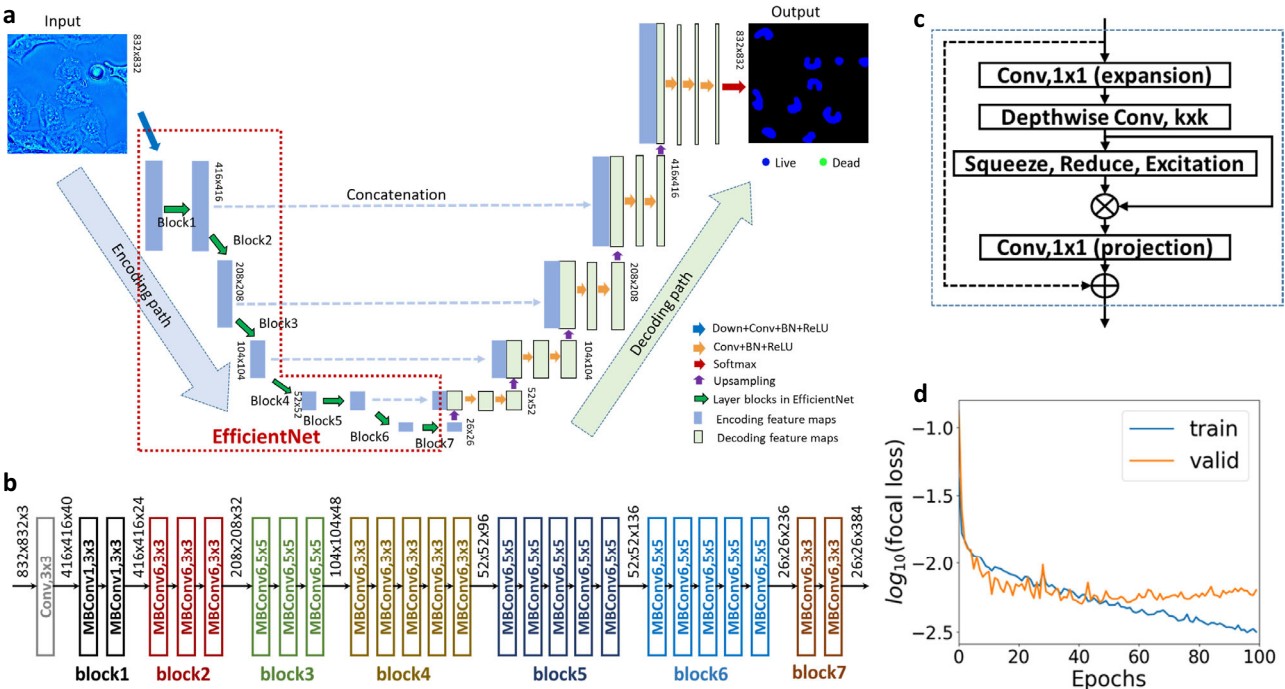

**Fig. 2 Principle of E-U-Net training. a** The E-U-Net. architecture includes an EfficientNet as the encoding path and five stages of decoding. The E-U-Net includes a Down+Conv+BN + ReLU block and 7 other blocks. The Down-Conv-BN-ReLU block represents a chain of down-sampling layer, convolutional layer, batch normalization layer, and ReLU layer. Similarly, the Conv+BN + ReLU is a chain of convolutional layer, batch normalization layer, and ReLU layer. **b** The network architecture of EfficientNet-B3. Different blocks are marked in different colors. They correspond to the layer blocks of EfficientNet in **a**. **c** The major layers inside the MBConvX module. X = 1 and X = 6 indicate the ReLU and ReLU6 are used in the module, respectively. The skip connection between the input and output of the module is not used in the first MBConvX module in each layer block. **d** Training and validation loss vs epochs plotted in the log scale.

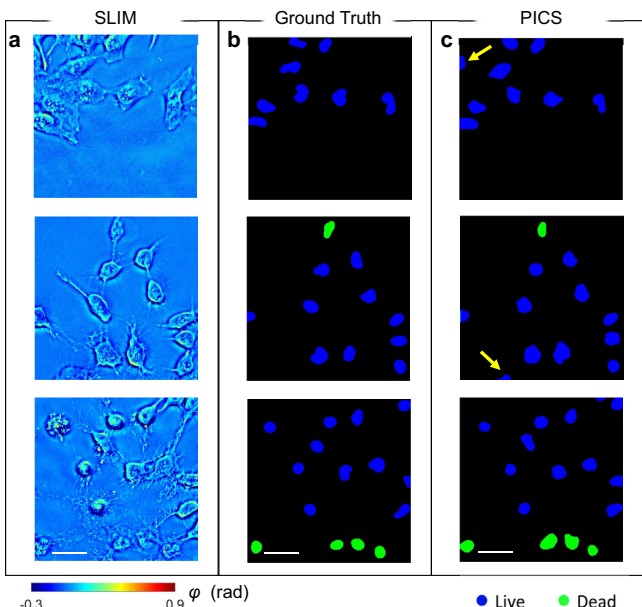

**Fig. 3 Results of E-U-Net on testing dataset. a** representative SLIM measurements of HeLa cells not used during training. **b** The ground truth for viability of frames corresponding to **a**. **c** The PICS prediction shows high level accuracy in segmenting the nuclear regions and inferring viability states. The arrows indicate the inconsistence between ground truth and PICS prediction caused by the cells located at the edge of the FOV are subject to inference error. The images are randomly selected from a combined dataset across 4 imaging experiments. Source data are provided as a Source Data file. Scale bars: 50 µm in space.

**Table 1 Evaluation of the E-U-Net performance.**

| | | Ground Truth | |
|---|---|---|---|
| | | Live (n = 1973) | Dead (n = 246) |
| PICS | Live | 98.8% | 2.4% |
| | Dead | 1.2% | 97.6% |
| Evaluation | Precision | 99.6% | 91.2% |
| | Recall | 98.8% | 97.6% |
| | F1 Score | 99.1% | 94.3% |

An object-based accuracy metric is used to estimate the deep learning prediction by comparing the dominant semantic label of HeLa cell nuclei with the ground truth. The entries of the confusion are normalized with respect to number of cells in each class.

1536 labeled SLIM images and 288 labeled SLIM images were used for network training and validation, respectively. The structure of EfficientNet-B7, training, and validation loss can be found in Fig. S3a, b, respectively. The trained E-U-net was finally applied to 288 unseen testing images to test the performance of dead/viability assay. The procedure of imaging, ground truth generation, and training was consistent with the previous experiments.

Figure 4a shows the time-lapse SLIM image of CHO cells measured at t = 0, 2, and 10 h after adding apoptosis reagent, and the corresponding viability map determined by fluorescence signal and PICS are plotted in Fig. 4b, c, respectively. In contrast to necrosis, the cell bodies became gradually fragmented during apoptosis. The visual comparison in Fig. 4 suggests that PICS yields good performance in extracting cell nucleus and predicting their viable state. Running an evaluation on individual cells, as

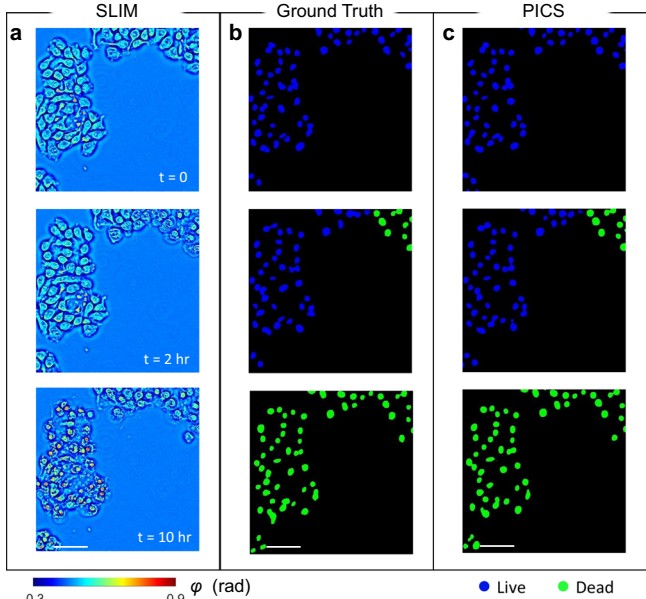

**Fig. 4 Results of PICS on adherent CHO cells. a** Time-lapse SLIM measurement of CHO cells measured at t = 0, 2, and 10 h. The data was not used in training or validation. **b** The ground truth for viability of frames corresponding to **a. c** The PICS prediction shows high level accuracy in segmenting the nuclear regions and inferring viability states. The images are randomly selected from a combined dataset across 4 imaging experiments. Scale bars: 50 μm in space.

**Table 2 Evaluation of the E-U-Net performance on CHO with apoptosis reagents.**

|  |  | Ground truth | |
|---|---|---|---|
|  |  | Live (n = 2071) | Dead (n = 6328) |
| PICS | Live | 90.1% | 1.7% |
|  | Dead | 9.9% | 98.3% |
| Evaluation | Precision | 94.6% | 96.8% |
|  | Recall | 90.1% | 98.3% |
|  | F1 Score | 92.3% | 97.5% |

The trained network yields high confidence in identifying live or apoptotic CHO cells. The entries of the confusion are normalized with respect to number of cells in each class.

shown in Table 2, the network gives an average F-1 score of 94.9%. Again, the inaccurate prediction is mainly caused by cells at the boundary of the FOV. We also found rare cases where cells show features of cells death at early stage[45–47], but it was identified as live by traditional fluorometric evaluation (for example, see Fig. S5 in the Supplemental Information). Furthermore, because most of the cells stay adherent, the PICS accuracy was not affected by cell confluence. The evaluation metrics at different confluence levels are included in the Supplemental Note 4.

**PICS on unlabeled HeLa cells.** Performing viability assay on unlabeled cells essentially circumvents the cell injury effect caused by exogenous staining and produces an unbiased evaluation. To demonstrate this feature on a different cell type, a fresh HeLa cell culture was prepared in a 6-well plate, transferred to the microscope stage, and maintained under room conditions. Half of the wells were mixed with viability assay reagents, where the viability

was determined by both PICS and fluorescence imaging. The remaining wells did not contain reagents, such that the viability of these cells was only evaluated by PICS. The procedure of cell preparation, staining, and microscope settings were consistent with the previous experiments. We took measurements every 30 min, and the entire experiment lasted for 12 h.

Figure 5a and c shows SLIM images of HeLa cells with and without fluorescent reagents at t = 0, 2.5, and 12 h, respectively, whereas the resulting PICS predictions are shown in Fig. 5b and d. Supplemental Video 1 shows a time-lapse SLIM measurement, PICS prediction, and standard live-dead assay based on fluorescent measurements. Supplemental Video 2 shows HeLa cells without reagents. As expected, the PICS method depicts the transition from live to dead state. In addition, the visual comparison from Fig. 5a–d suggests that HeLa cells with viability stains in the media appear smaller in size, and more rapidly enter the injured state, as compared to their label-free counterparts. Using TrackMate[48], an ImageJ plugin, we were able to extract the trajectory of individual cells and track their morphology over time. As a result, the cell nucleus, area, and dry mass at each moment in time can be obtained by integrating the pixel value over the segmented area in the PICS prediction and SLIM image, respectively. We successfully tracked 57 labeled and 34 unlabeled HeLa cells. Figure 5e, f shows the area and dry mass change (mean ± standard error), where the values are normalized with respect to the one at t = 0. Our results of tracking agree with the physiological description[49,50], and are consistent with previously reported experimental validations[46,51]. However, the short swelling time in the reagent-treated cells suggests the toxicity of the chemical compounds would potentially accelerate the pace of cell death. Running two-sample t-tests, we found a significant difference in cell nuclear areas between the labeled and unlabeled cells, during the interval t = 2 and t = 7 h (p < 0.05). Similarly, cell dry mass showed significant differences between the two groups during the time interval t = 2 and t = 5 h (p < 0.05). In this study, we focus on optimizing the PICS performance in classifying live/dead markers at the cellular level. At the pixel level, the trained network can reveal the cell shape change, but its performance in capturing the nucleus shape and area is limited, which makes the current approach subject to segmentation error. This is largely due to the low contrast between the nucleus boundary and cytoplasm in injured cells.

Although the effect of the fluorescent dye itself on the optical properties of the cell at the imaging wavelength is negligible[52–55], training on images of tagged cells may potentially alter the cell death mechanism and introduce bias when optimizing the E-U-Net. To investigate this potential concern, we performed a set of experiments where the unlabeled cells were imaged first by SLIM, then tagged and imaged by fluorescence for ground truth. As described in Supplemental Note 4, we found that the performance of PICS, in this case, was consistent with the results shown in Figs. 3 and 4, where SLIM was applied to tagged cells. The data indicated that the live and dead cells were classified with 99% and 97% sensitivity, respectively, suggesting that the proposed live-dead assay method can be used efficiently on cells that were never labeled. Of course, SLIM imaging of already stained cells, followed by fluorescence imaging, is a more practical workflow, as the input-ground truth image pairs can be collected continuously. On the other hand, training on unlabeled cells allows us to achieve the true label-free assay which is most valuable in applications.

## Discussion

We demonstrated PICS as a method for high-speed, label-free, unbiased viability assessment of adherent cells. This approach utilizes quantitative phase imaging to record high-resolution

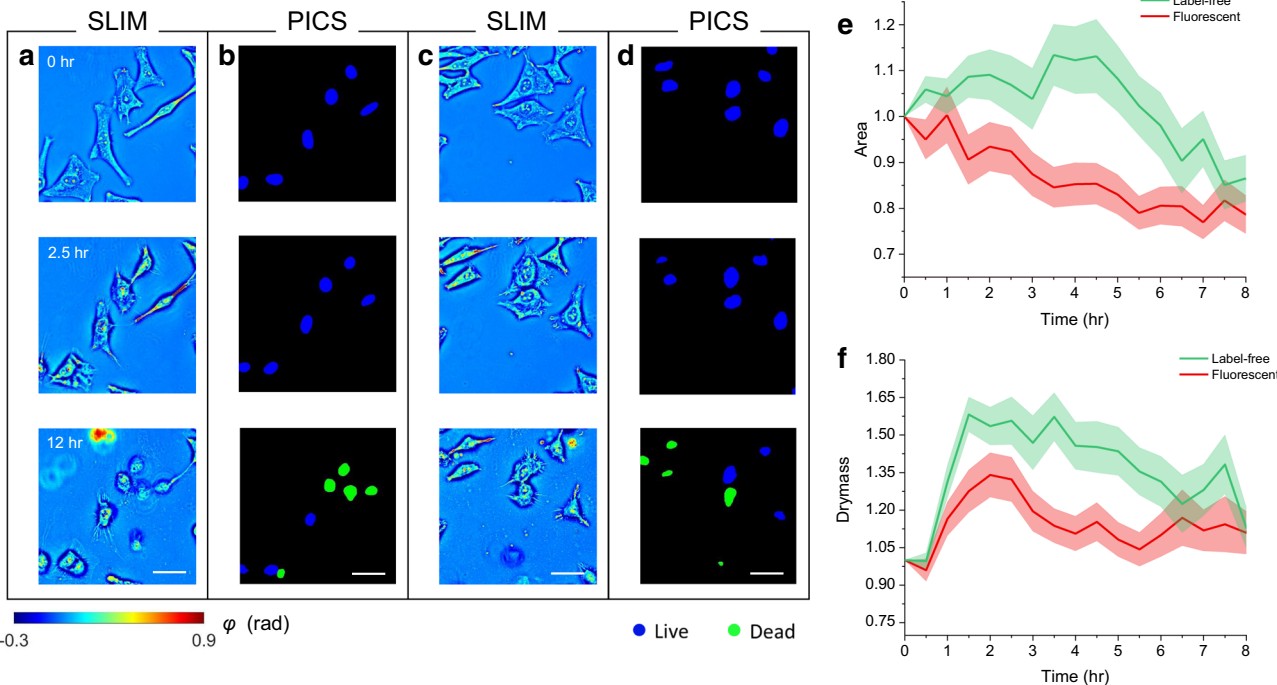

**Fig. 5 Viability of HeLa cells with and without reagent stains. a**. SLIM images of cells recorded at 0, 2.5 and 12 h after staining. **b**. The PICS prediction associated with the frames in **a**. **c**. SLIM images of unstained HeLa cells measured at same time points as **a**. **d**. The corresponding PICS prediction associated with the frames in **c**. **e**. Relative cell nuclear area change of tracked cells. The central line represents the average, and the shaded region indicates the standard error. **f**. Relative cell nucleus dry mass change. The central line represents the average, and the shaded region indicates the standard error. Source data are provided as a Source Data file. Scale bars: 50 μm in space.

morphological structures of unstained cells, combined with deep learning techniques to extract intrinsic viability markers. Tested on HeLa and CHO adherent cultures, our optimized E-U-Net method reports outstanding accuracy of 96.7% and 94.9% in segmenting the cell nuclei and classifying their viability state. In Supplemental Note 5, we compared the E-U-Net accuracy with the outcomes from other networks or training strategies. By integrating the trained network on NVIDIA graphic processing units, the proposed label-free method enables real-time acquisition and viability prediction (see Supplemental Video 3 for a demonstration). One SLIM measurement and deep learning prediction take ~100 ms, which is approximately 8 times faster than the acquisition time required for fluorescence imaging with the same camera. Of course, the cell staining process itself takes time, approximately 15 min in our case. The real-time in situ feedback is particularly useful in investigating viability state and growth kinetics in cells, bacteria, and samples in vivo over extended periods of time[56–59]. In addition, results suggest that PICS rules out the adverse effect on cell function caused by the exogenous staining, which is beneficial for the unbiased assessment of cellular activity over a long time (e.g., many days). Of course, this approach can be applied to other cell types and cell death mechanisms.

Prior studies typically tracked QPI parameters associated with individual cells over time to identify morphological features correlated with cell death[45,46,51]. In contrast, our approach provides a real-time classification of cells based on single frames, which is a much more challenging and rewarding task. Compared to these previous studies, our PICS method avoids intermediate steps of feature extraction, manual annotation, and separate algorithms for training & cell classification. We employ a single DNN architecture with direct QPI measurement as input, and the prediction accuracy is significantly improved over the previously reported data[47]. The labels output by the network can be used to create binary masks,

which in turn yield dry mass information from the input data. The accuracy of these measurements depends on the segmentation process. Thus, we anticipate that future studies will optimize further the segmentation algorithms to yield high-accuracy dry mass measurements over long periods of time.

Label-free imaging methods are valuable for studying biological samples without destructive fixation or staining. For example, by employing infrared spectroscopy, the bond-selective transient phase imaging measures molecular information associated with lipid droplets and nucleic acids[60]. In addition, harmonic optical tomography can be integrated into an existing QPI system to report specifically on non-centrosymmetric structures[61]. These additional chemical signatures would potentially enhance effective learning and produce more biophysical information. We anticipate that the PICS method will provide high-throughput cell screening for a variety of applications, ranging from basic research to therapeutic development and protein production in cell reactors[11]. Because SLIM can be implemented as an upgrade module onto an existing microscope and integrates seamlessly with fluorescence, one can implement this label-free viability assay with ease.

## Methods

**Cell preparation**. HeLa cervical cancer cells (ATCC CCL-2™) and Chinese hamster ovary (CHO-K1 ATCC CCL-61™) cells were purchased from ATCC and kept frozen in liquid nitrogen. Before the experiments, we thawed and cultured the cells into a T75 flask in Dulbecco's Modified Eagle Medium (DMEM with low glucose) containing 10% fetal bovine serum (FBS) and incubated in 37 °C with 5% $CO_2$. As the cells reach 70% confluence, the flask was washed thoroughly with phosphate-buffered saline (PBS) and trypsinized with 3 mL of 0.25% (w/v) Trypsin EDTA for three minutes. When the cell starts to detach, the cells were suspended in 5 mL DMEM and passaged onto a glass-bottom 6 well plate to grow. To evaluate the effect of confluency on PICS performance, CHO cells were plated in three different confluency levels: high (60,000 cells), medium (30,000 cells), and low (15000 cells). HeLa and CHO cells were then imaged after two days.

**SLIM imaging**. The SLIM optical setup is shown in Fig. 1a. In brief, the microscope is built upon an inverted phase-contrast microscope with a SLIM module (CellVista SLIM Pro; Phi Optics) attached to the output port. Inside the module, a spatial light modulator (Meadowlark Optics) is placed at the system pupil plane via a Fourier transform lens to constantly modulate the phase delay between the scattered and incident light. By recording four intensity images with phase shifts of 0, $\pi/2$, $\pi$, and $3\pi/2$, a quantitative phase map, $\varphi$, can be computed by combining the 4 acquired frames in real-time.

For both SLIM and fluorescence imaging, cultured cells were measured by a 40× objective, and the images were recorded by a CMOS camera (ORCA-Flash 4.0; Hamamatsu) with a pixel size of 6.5 μm. For each sample, we randomly selected a cellular region approximately $800 \times 800$ μm$^2$ to be measured by SLIM and fluorescence microscopy (NucBlue and NucGreen). The acquisition time of each SLIM and fluorescent measurements are 50 ms and 400 ms, respectively, and the scanning across all 6 wells takes roughly 4.3 min, where the delay is caused by mechanical translation of the motorized stage. For deep learning training and predicting, the recorded SLIM images were downsampled by a factor of 2. This step saves computational cost and does not sacrifice information content. We would like to point out that the acquisition of the fluorescence data is needed only for the training stage. For real-time interference, our acquisition is up to 15 frames per second for SLIM images, while the inference takees place in parallel.

**E-U-Net architecture**. The E-U-Net is a U-Net-like fully convolutional neural network that performs an efficient end-to-end mapping from SLIM images to the corresponding probability maps, from which the desired segmentation maps are determined by the use of a softmax decision rule. Different from conventional U-Nets, the E-U-Net uses a more efficient network architecture, EfficientNet[37], for feature extraction in the encoding path. Here, EfficientNets refers to a family of deep convolutional neural networks that possess a powerful capacity of feature extraction but require much fewer network parameters compared to other state-of-the-art network architectures, such VGG-Net, ResNet, Mask R-CNN, etc. The EfficientNet family includes eight network architectures, EfficientNet-B0 to EfficientNetB7, with an increasing network complexity. EfficientNet-B3 and EfficientNet-B7 were selected for training E-U-Net on HeLa cell images and CHO cell images, respectively, considering they yield the most accurate segmentation performance on the validation set among all the eight EfficientNets. See Supplemental Note 2 and Fig. 2b, c for more details about the EfficientNet-B3 and EfficientNet-B7.

**Loss function and network training**. Given a set of $B$ training images of $M \times N$ pixels and their corresponding ground truth semantic segmentation maps, loss function used for network training is defined as the combination of focal loss[62] and dice loss[63]:

$$L_{Focal\_loss} = -\frac{1}{B}\sum_{i=1}^{B}\frac{1}{MN}\sum_{x\in\Omega}[1-y_i(x)^T p_i(x)]^\gamma y_i(x)^T \log_2 p_i(x), \quad (1)$$

$$L_{Dice\_loss} = 1 - \frac{1}{3}\sum_{c=0}^{2}\frac{2TP_c}{2TP + FP_c + FN_c} \quad (2)$$

$$L_{combined} = \alpha L_{Focal\_loss} + \beta L_{Dice\_loss} \quad (3)$$

In the focal loss $L_{Focal\_loss}$, $\Omega = \{(1, 1), (1, 2), ..., (M, N)\}$ is the set of spatial locations of all the pixels in a label map. $y_i(x) \in \{[1, 0, 0]^T, [0, 1, 0]^T, [0, 0, 1]^T\}$ represents the ground-truth label of the pixel $x$ related to the $i^{th}$ training sample, and the three one-hot vectors correspond to the live, dead and, background classes, respectively. Accordingly, the probability vector $\mathbf{P}_i(x) \in \mathbb{R}^3$ represents the corresponding predicted probabilities belonging to the three classes. $[1 - y_i(x)^T p_i(x)]^\gamma$ is a classification error-related weight that reduces the relative cross-entropy $y_i(x)^T \log_2 p_i(x)$ for well-classified pixels, putting more focus on hard, misclassified pixels. In this study, $\gamma$ was set to be the default value of 2 as suggested in Ref. [62]. As the dice loss $L_{Dice\_loss}$, the $TP_c$, $FP_c$, and $FN_c$ are the number of true positives, that of false positives, and that of false negatives, respectively, related to all pixels of viability class $c \in \{0, 1, 2\}$ in the B images. Here, $c = 0$, 1, and 2 correspond to the live, dead, and background classes, respectively. In the combined loss function, $\alpha, \beta \in \{0, 1\}$ are two indicators that control whether to use focal loss and dice loss in the training process, respectively. In this study, $\alpha$ and $\beta$ were set to [1, 0] and [1, 1] for training the E-U-Nets on the HeLa cell dataset and CHO cell dataset, respectively. The choices of $[\alpha, \beta]$ were determined by the segmentation performance of the trained E-U-Net on the validation set.

The E-U-Net was trained with randomly cropped patches of $512 \times 512$ pixels drawn from the training set by minimizing the loss function defined above with an Adam optimizer[43]. In regard to Adam optimizer, the exponential decay rates for 1$^{st}$ and 2$^{nd}$ moment estimates were set to 0.9 and 0.999, respectively; a small constant $\varepsilon$ for numerical stability was set to $10^{-7}$. The batch sizes were set to 14 and 4 for training the E-U-nets on the HeLa cell images and CHO cell images, respectively. The learning rate was initially set to $5 \times 10^{-4}$. At the end of each epoch, the loss of a being-trained E-U-Net was computed on the whole validation set. When the validation loss did not decrease for 10 training epochs, the learning rate was multiplied by a factor of 0.8. This validation loss-aware learning rate decaying strategy benefits for mitigating the overfitting issue that commonly occurs in deep neural network training. Furthermore, data augmentation techniques, such as random cropping, flipping, shifting, and random noise and brightness adding, etc.,

were employed to augment training samples on the fly for further reducing the overfitting risk. The E-U-Net was trained for 100 epochs. The parameter weights that yield the lowest validation loss were selected and subsequently used for model testing and further model investigation.

The E-U-Net was implemented using the Python programming language with libraries including Python 3.6 and Tensorflow 1.14. The model training, validation, and testing were performed on an NVIDIA Tesla V100 GPU of 32 GB VRAM.

**Reporting summary**. Further information on research design is available in the Nature Research Reporting Summary linked to this article.

## Data availability

Source data producing the graphs in the article are provided with this paper. In addition, an example set of the SLIM images and corresponding viability maps used in this study is deposited in the GitHub repository "shenghh2015/label-free-viability-assay" [https://github.com/shenghh2015/label-free-viability-assay]. The complete dataset is not deposited due to its massive size, but is available from the corresponding author upon request. Requests will be answered within two weeks. Source data are provided with this paper.

## Code availability

The MATLAB script to generate semantic segmentation maps, along with representative input images is provided as Supplementary Software. The trained E-U-Net models along with sample testing images are available for download at https://github.com/shenghh2020/label-free-viability-assay.

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

## Acknowledgements

National Science Foundation (0939511, 1353368), G.P. National Institute of Health (R01CA238191, R01GM129709), G.P. National Institute of Health – Tissue Micro-environment (TiMe) training program (T32EB019944), C.H.

## Author contributions

G.P. and C.H. conceived the idea. C.H. designed the experiment, collected data, generated semantic segmentation maps, performed deep learning testing, and conducted data analysis. S.H. led the deep learning training and validation. Y.L. prepared cell cultures. Y.H. performed software integration. E.M.K prepared the script for PICS evaluation. C.H. and S.H wrote the manuscript with input from all authors. M.A.A. and H.L. supervised the deep learning. G.P. supervised the project.

## Competing interests

G.P. has financial interest in Phi Optics, a company developing quantitative phase imaging technology for materials and life science applications, which, however, did not sponsor the research. C.H., S.H., Y.H., M.A., and G.P. have disclosed the invention to the Office of Technology Management at the University of Illinois at Urbana-Champaign for patent applications. The remaining authors declare no competing interests.
