## [Peer Review File · Nature Communications]

Reviewers' Comments:

Reviewer #1:

Remarks to the Author:

In the article called „Live-dead assay on unlabeled cells using phase imaging with computational specificity“ Chenfei Hu and co-authors introduce a method for analysis of cell viability using time-lapse tracking of QPI-measured cell dry mass and projected area.

Quantitative phase imaging and cell death analysis are rapidly developing areas in the field of light microscopy and cell biology, respectively. Label-free quantitative phase imaging cell death analysis has undoubtedly promising potential in high-throughput automatic approaches in the field of drug discovery and cytotoxicity measurement. Nevertheless, approaches used in this article are not novel and showed no added value. QPI for cell death detection was reported by Pavillon et al. (<https://journals.plos.org/plosone/article?id=10.1371/journal.pone.0030912>) and Rommel et al. (<https://www.spiedigitallibrary.org/conference-proceedings-of-spie/9129/912940/Multimodal-label-free-in-vitro-toxicity-testing-with-digital-holographic/10.1117/12.2054374.short?SSO=1>). More recently, Vicar et. al. distinguished apoptotic and lytic cell death subroutines using time-course QPI micrographs (<https://www.nature.com/articles/s41598-020-58474-w>).

Chenfei Hu's article shows many similar patterns with the above-mentioned articles. This work also does not reach the experimental quality of the above-mentioned works.

To strengthen the conclusions, authors should pay more attention to the biological part of the experiment. It seems to be heavily underdeveloped in this article. Here are some examples from the Results section:

„To exclude the effect of necrosis, the cells were maintained in regular incubation condition throughout the experiment“.

The induction of lytic cell death (in this article referred to as „necrosis“) cannot be eliminated by placing the cells in the incubator with standardized conditions. Lytic cell death is mediated by highly orchestrated molecular pathways which take place even in the standardized cultivation environment. How do the authors know that the mentioned conditions will not lead to necrosis? Did they measure the pH inside the cultivation chamber? Were some lytic cell death markers measured (DAMPs, p-MLKL, GSDMD,..)?

The authors claim that viability measuring reagents decrease viability (Abstract and Results section), but for the fluorescent measurement, they used 400 ms exposition which is, in the case of UV light used for DAPI excitation, in itself toxic. Authors should verify their conclusion using some classic viability tests, such as MTT or XTT with no excitation light to assess the decrease in viability caused by reagents.

A formulation such as „two drops of cell-viability reagent were added“ should be avoided in scientific literature. The exact volume or final concentration of the reagent should be provided.

From the image analysis point of view, the proposed method is simple and straightforward, but achieves very promising results. The proposed method is very similar to paper from the same research group <https://doi.org/10.1038/s41467-020-20062-x>, where they predict cell fluorescently stained cell compartments instead of living/dead cells; thus, the proposed method is not very innovative and it is only a new application of the same thing. The article is completely missing a literature review of deep learning methods predicting fluorescence staining from label-free images; thus, it is not clear what is novel in this paper.

Also, the proposed method is not sufficiently tested and evaluated. For example, I would appreciate a comparison of used pretrained E-U-Net with not-pretrained network and with the standard U-Net architecture. Similarly, I miss the comparison of the used loss function (mixture of standard dice loss and focal loss) with standard dice loss or weighted cross-entropy. For clarity, I would move loss function description from supplementary information into the manuscript. Is the prediction of thresholded fluorescence staining better than the direct prediction of raw fluorescence image?

The description of „object-based evaluation metric“ is missing.

How was the training/validation/testing splitting done? Is the testing dataset sample from all „data batches“ or does the testing dataset come from different „data batch“ („We repeated this experiment for four times to capture the variability among different batches.“)? Testing should be done on data from a different batch; otherwise, it can falsely increase the accuracy of the proposed method.

For these reasons, I do not recommend this article for publication.

Reviewer #2:

Remarks to the Author:

Review of NCOMMS-21-12345, "Live-dead assay on unlabeled cells using phase imaging with computational specificity," by C. Hu et al.

This manuscript presents application of quantitative phase imaging to evaluating cell viability based on a deep learning approach, termed computation specificity. This work is motivated by suggestion that viability reagents decrease cell viability. Offering a non-invasive alternative could have significant benefits.

This is a nice demonstration of the utility of QPI and illustrates the powerful combination with deep learning techniques. One requirement for this type of algorithm is a large number of input data. Some discussion of how long it took to acquire these data would be a helpful inclusion. The methods section gives the exposure time but given the large number of cells examined, there must have been some variation in what time the cells were imaged vs. time after exposure. Claiming this as a high-throughput method, as done in the discussion, should be backed up with some details on imaging throughput.

The illustration of loss vs number of epochs isn't very illuminating. The loss appears to drop to <2% immediately and then doesn't decrease much. Perhaps a better scaling (log ?) would highlight this better. Otherwise it appears that there is not much point in training more than a bit. The results appear quite good but a little more examination of how and why the algorithm fails would be a good inclusion. That is Fig 3b,c only show correct classification but readers may wish to understand the limitations. It is discussed but only briefly in the results section. Some discussion on how the pixel-wise evaluation relates to the dominant semantic label would also help.

The experiments with CHO cells are very similar to a recent publication from Eldridge et al (J. Biophotonics 2018) where cells were treated with staurosporine and examined with QPI. This earlier work also compared with a viability assay in the form a fret based apoptosis sensor. Some comparison with this earlier work would better place the current paper in perspective. The E-U-Net performance (Table 2) shows that the algorithm only correctly identifies 90% of live cells as live and nearly 10% as dead. Some discussion of this aspect, its potential impact on utility, and possible pathways to improve performance would also strengthen the paper.

The results showing that cell staining results in a decrease in area and dry mass are interesting and reinforce the need for this label-free assay. Some discussion of the increase in these parameters, which occurs for both treated and untreated, should also be included.

In summary, this paper presents an advance in using QPI imaging for analyzing cell features. The advance here is the use of deep learning to identify cell viability. The study is well presented and the manuscript is well written. I recommend publication after the above minor revisions.

Reviewer #3:

Remarks to the Author:

In this article, the authors presented a label-free cell viability assessment approach, in which a deep neural network was trained to predict the segmentation maps of the live/dead cells from the quantitative phase images measured using spatial light interference microscopy (SLIM). This approach was demonstrated by imaging live adherent CHO and HeLa cells over time. By comparing the network predictions with the ground truth, a cell viability prediction accuracy of 95% was achieved. The cell nucleus area and dry mass were further evaluated based on the segmentation maps.

The work is timely, very exciting, clearly organized and well presented. Furthermore it would be of broad interest for the interdisciplinary readership of Nat Comm. Therefore, I recommend its publication after the following comments are addressed by the authors.

The presented neural network was trained and tested with labelled cells, therefore the reported performance of PICS method was not truly label-free.

In this work, the authors added the fluorescent reagents to the cell culture before acquiring SLIM images. There are two issues that need to be considered here:

1. The viability stains can potentially change the cell morphology. This was also indicated in the article, that the labeled HeLa cells appear smaller in size compared to their label-free counterparts. Therefore, the network model, which was optimized with the images of labeled cells, perhaps may not generate entirely reliable predictions of unlabeled cells (Fig. 5d).

2. The viability stains might alter the cells' optical properties thus introduce additional information to the captured SLIM images. This additional information may help the network differentiate live/dead cells, but is not available for the unlabeled cells during blind testing. The reported accuracy of 95% was calibrated with images of labelled cells, which does not represent the general performance of this method on unlabeled cells.

As a suggestion, to better demonstrate the feasibility of the method, the phase images could perhaps be captured before adding the viability reagents. Then quickly perform viability labeling and capture the fluorescence images, which could serve as ground truth. Based on this, the accuracy and other metrics could be calculated. Since the cells are already plated, the alignment should not be a problem.

Reviewer #1 (Remarks to the Author):

In the article called “Live-dead assay on unlabeled cells using phase imaging with computational specificity” Chenfei Hu and co-authors introduce a method for analysis of cell viability using time-lapse tracking of QPI-measured cell dry mass and projected area.

We thank the reviewer for reading through our manuscript and providing constructive comments. These comments are helpful in clarifying the novelty and significance of the reported study. Below, we addressed the reviewer’s concerns individually and revised our manuscript correspondingly.

Quantitative phase imaging and cell death analysis are rapidly developing areas in the field of light microscopy and cell biology, respectively. Label-free quantitative phase imaging cell death analysis has undoubtedly promising potential in high-throughput automatic approaches in the field of drug discovery and cytotoxicity measurement. Nevertheless, approaches used in this article are not novel and showed no added value. QPI for cell death detection was reported by Pavillon et al. (<https://journals.plos.org/plosone/article?id=10.1371/journal.pone.0030912>) and Rommel et al. (<https://www.spiedigitallibrary.org/conference-proceedings-of-spie/9129/912940/Multimodal-label-free-in-vitro-toxicity-testing-with-digital-holographic/10.1117/12.2054374.short?SSO=1>). More recently, Vicar et. al. distinguished apoptotic and lytic cell death subroutines using time-course QPI micrographs (<https://www.nature.com/articles/s41598-020-58474-w>). Chenfei Hu's article shows many similar patterns with the above-mentioned articles. This work also does not reach the experimental quality of the above-mentioned works.

We thank the reviewer for pointing our attention to these references that we are familiar with, but failed to cite. We now cite all these relevant papers. What all these prior studies have in common is the idea of tracking QPI parameters associated with individual cells over time, thus, achieving a score in “tens of minutes”. By contrast, our approach provides real-time classification of cells based on single frames, which is much more challenging and, we hope, more useful for biological studies.

Compared to these previous studies [1], our PICS method avoids intermediate steps of feature extraction, manual annotation, and separate algorithms for training & cell classification. Our method employs a single deep neural network (DNN) architecture, and the prediction accuracy is significantly improved, comparing with the previously reported data. This DNN used direct QPI images as input, and the ground truth was produced by automated straightforward image processing script.

In the revised Discussion section, we now clarify the distinction with respect to these previous studies on pp.11.

To strengthen the conclusions, authors should pay more attention to the biological part of the experiment. It seems to be heavily underdeveloped in this article. Here are some examples from the Results section: “To exclude the effect of necrosis, the cells were maintained in regular incubation condition throughout the experiment”. The induction of lytic cell death (in this article referred to as “necrosis”) cannot be eliminated by placing the cells in the incubator with standardized conditions. Lytic cell death is mediated by highly orchestrated molecular pathways which take place even in the standardized cultivation environment. How do the authors know that the mentioned conditions will not lead to necrosis? Did they measure the pH inside the cultivation chamber? Were some lytic cell death markers measured (DAMPs, p-MLKL, GSDMD,...)?

This is an important point that we now explain better in the revised manuscript, as follows. Before performing experiments on CHO cells, a preliminary study was conducted to evaluate the efficacy of staurosporine on introducing cell apoptosis. We prepared live cell cultures and split them into the two groups. 1 μ M of staurosporine was added into the medium of the experimental group, whereas the others were kept intact as control. Both control and experimental cells were measured with SLIM for 10 hours under regular incubation condition (37 °C and 5% concentration of CO₂). Figure 1a and 1b show the QPI images of experimental and control cells measured at t = 0.5, 6.5, 7 and 10 hours, respectively. Throughout the time-course, the untreated cells remained attached to the petri-dish. Moreover, as indicated by the yellow arrow, the control cells divided at t = 6.5 hr. In contrast, cells treated with staurosporine presented drastically different characteristics, where the cell

volume decreased, and membrane ruptured or became detached. This preliminary result suggests that, under the regular incubation condition, the cells did not suffer from lytic cell death.

In the revised manuscript, we now added this clarification to the Supplemental Information on pp.4.

Figure 1. Time-lapse SLIM recording of cells with (a) and without (b) staurosporine that introduces cell apoptosis, under regular incubation condition. For the control group, the cells continued growing and dividing without signs of cell death, which ruled out the existence of lytic cell death.

The authors claim that viability measuring reagents decrease viability (Abstract and Results section), but for the fluorescent measurement, they used 400 ms exposition which is, in the case of UV light used for DAPI excitation, in itself toxic. Authors should verify their conclusion using some classic viability tests, such as MTT or XTT with no excitation light to assess the decrease in viability caused by reagents.

The reviewer is correct that fluorescence-based assay kits are not ideal either, which is part of the motivation for our label-free method. MTT or XTT assay estimates the viability of a cell population by measuring the optical absorbance caused by formazan concentration due to alteration in mitochondrial activity [2-4]. Although this assay provides sensitive viability outcome, it does not give an estimation for individual cells, and it is against the goal of our

study. Furthermore, to the best of our knowledge, this assay protocol typically requires 2-4 hours of MTT labeling, and even longer time for solubilizing formazan [4]. In revision, we added this comparison to the introduction (pp. 2).

On the other hand, trypan blue exclusion assay and fluorescence-based cell viability are commonly used methods for fast viability assessment at cellular level since 1970s [5]. Although the fluorometric assay is proven to be less toxic, the photo-toxicity associated with the exogenous fluorescent markers are inevitable [5]. And our results in Figure 5 (nucleus morphologies of fluorescent-labeled and unlabeled cells vs. time) suggest this potential influence.

A formulation such as “two drops of cell-viability reagent were added” should be avoided in scientific literature. The exact volume or final concentration of the reagent should be provided.

We thank the reviewer for catching this qualitative language. Unfortunately, precise information about the product, such as concentration or reagent volume, was withheld by the manufacturer and not available to the users (<https://www.thermofisher.com/us/en/home/references/protocols/cell-and-tissue-analysis/protocols/readyprobes-cell-viability-imaging-kit-blue-green.html>). To mitigate this issue, we measured the volume of each drop and arrived at a value of 20 μ L. In the revised manuscript, we specify that “40 μ L of each cell-viability-assay reagent (ReadyProbes Cells Viability Imaging Kit, Thermofisher) was added into 1 ml growth media”.

From the image analysis point of view, the proposed method is simple and straightforward, but achieves very promising results. The proposed method is very similar to paper from the same research group <https://doi.org/10.1038/s41467-020-20062-x>, where they predict cell fluorescently stained cell compartments instead of living/dead cells; thus, the proposed method is not very innovative and it is only a new application of the same thing.

We thank the review for acknowledging that our results are “very promising”. Please allow us to clarify the novelty of the study. First, existing methods for cell viability assays involve converting a population of cells into a colored product [6]. These conventional methods either require certain time of sample preparation and introduce toxicity, which prevent continuous monitoring the reaction of same cells over a long time. We have learned that large Pharma companies still use the “trypan blue” dye to assess viability in their cell reactors, despite the fact that this chemical is toxic and, within minutes, will end up killing all the cells. Very few studies have been reported for reliable instantaneous cell viability screening without labeling.

Second, although the workflow of image acquisition resembles the one from our previous study [7], the training, results, and implications, along with the respective challenges, are very different. The end goal of our studies is not to simply predict fluorescence stains, but to extract the biologically specific information about the samples directly from quantitative phase imaging. Since such specificity was usually obtained from fluorescence microscopy and different dyes, we used them to generate the ground truth data. The use of fluorescence signals in our work is equivalent to manual annotation/labeling in standard deep learning applications. Fortunately, due to the integration of our QPI module with the fluorescence channels, we avoid the manual annotation and create ground truth data automatically with ease.

Third, in our previous work, we were able to feed the fluorescence images directly into training only because both channels (DAPI and DiI) worked independently and marked different regions of interest. In this work, however, a combination of DAPI and FITC channels is required to tell the viability of cells (as described in the Supplemental Information Section 1). As a result, we applied image analysis techniques on both channels to create live-dead segmentation masks and trained the deep neural network using these masks.

The article is completely missing a literature review of deep learning methods predicting fluorescence staining from label-free images; thus, it is not clear what is novel in this paper.

To our knowledge, the idea of using deep learning to predict fluorescence maps has been brought up since 2018. Researchers from Google presented "in silico labeling" (ISL), a deep learning based approach that can predict fluorescent labels from transmitted-light (bright field and phase contrast) images of unlabeled samples [8]. In the same year, researchers from the Allen Institute showed that individual subcellular structure such as DNA, cell membrane, and mitochondria can be obtained computationally from bright-field images [9]. In the field of QPI, researchers from UCLA presented in 2019 that deep learning methods can apply histological stains on both QPI images [10] and autofluorescence images [11] of unlabeled tissue sections. Researches from Tel Aviv University demonstrated in the same year that deep learning can help "virtually stain" holographic microscopy images of human sperm cells for fertility evaluation [12].

The development of all these methods involves the use of stained samples at some point, since annotated data is indispensable from supervised learning. However, none of these works aim only to "predict the stains", they all served as novel alternatives to biological tasks and more importantly, contributed to answering the fundamental question about the information capacity of label-free imaging.

Our proposed method was inspired by all the previous works and continued exploring the information contained within QPI. We combined recently-developed, state-of-the-art neural networks [13], advanced training strategies (detailed in Methods) with SLIM to assay the viability of live mammalian cells. Our results showed that PICS provides a fast and convenient approach for cell live/dead assay and can potentially be used broadly to further investigate the link between cell dynamics and viability status.

In the original manuscript, we have cited a number of these studies about image transformation with deep learning, specifically for QPI (pp. 2). We now discuss more studies particularly about translating label-free images to fluorescent results to the revision, to make this point clearer (pp. 2).

Also, the proposed method is not sufficiently tested and evaluated. For example, I would appreciate

a comparison of used pretrained E-U-Net with not-pretrained network and with the standard U-Net architecture.

We thank the reviewer for the comments and suggestions. As suggested, we have added comparison experiments to further test and evaluate the E-U-net method.

We have attempted to compare cell viability prediction performance under various network architecture settings. We would like to clarify that in our previous E-U-net training procedure, we did not pre-train the whole E-U-net in the network initialization; instead we used the weights of a pre-trained EfficientNet [14] to initialize the encoder of the E-U-net but randomly initialize the decoder part of the E-U-net. Thus, here, we assume the “used pre-trained E-U-net” in the reviewer’s comment refers to an E-U-net trained with a pre-trained EfficientNet, and an “E-U-net without pre-trained network” refers to an E-U-net trained from scratch. Based on such interpretation, we compared three network settings: 1) an E-U-net trained by use of a pre-trained EfficientNet; 2) an E-U-net trained from scratch; and 3) a standard U-net [15] trained from scratch.

In these additional experiments, the U-net architecture employed was a standard U-net [15], with the exception that batch normalization layers were placed after each convolutional layer to facilitate the network training. EfficientNet-B0 was employed in the E-U-nets to make sure that the network size of E-U-net (7.8 million of parameters) approximately matched that of a standard U-net (7.85 million of parameters). A combined loss that comprised focal and dice losses (denoted as dice+focal loss) was used for network training. Other training settings were consistent with how the E-U-net was trained, as described in the manuscript. After the networks were trained with training and validation data from HeLa cell datasets and CHO cell datasets, they were tested on the testing data from the two datasets, respectively. The average pixel-wise F1 scores over the live, dead and background classes were computed to evaluate the performance of the trained networks, as reported in Table 1. It can be observed from the table that, on both the two testing datasets, the average F1 scores corresponding to an E-U-net are much higher than those corresponding to a standard U-net when both of them were trained from scratch. Furthermore, as expected, an E-U-net trained with a pre-trained EfficientNet achieves a better performance than the one trained from scratch. These

results demonstrate the effectiveness of the E-U-net architecture and the transfer learning techniques in training a deep neural network for pixel-wise cell viability prediction. We have also added these comparison experimental results in the revised Supplementary Information Section 5.

Network	E-U-Net (pre-trained)	E-U-Net (scratch)	Standard U-Net (scratch)
HeLa	80.6%	75.0%	72.0%
CHO	85.7%	80.4%	76.0

Table 1. Average F1 scores related to E-U-nets trained with a pre-trained EfficientNet-B0, E-U-nets trained from scratch, and standard U-nets trained from scratch, respectively.

Similarly, I miss the comparison of the used loss function (mixture of standard dice loss and focal loss) with standard dice loss or weighted cross-entropy. For clarity, I would move loss function description from supplementary information into the manuscript.

Following reviewer’s comment, we compared the average pixel-wise F1 scores corresponding to E-U-nets trained with various loss functions, including a dice+focal loss, a standard focal loss, a standard dice loss, and a weighted cross entropy (WCE) loss. To be consistent with the network settings in the manuscript, a pre-trained EfficientNet-B3 and a pre-trained EfficientNet-B7 were employed for training the E-U-nets on the HeLa cell dataset and CHO cell datasets, respectively. The class weights related to live, dead, and background classes in the weighted cross entropy loss were set to [0.17, 2.82, 0.012] and [2.32, 0.654, 0.027] for the network training on the HeLa cell dataset and CHO cell datasets, respectively. In each of the weight cross entropy losses, the average of weights over the three classes is 1, and the weights related to each class were inversely proportional to the percentages of pixels from each class in the HeLa cell and CHO cell training datasets: [6.7%, 0.4%, 92.9%] and [1.1%, 3.9%, 95%], respectively. Other network training settings were consistent with how the E-U-net was trained as described in the manuscript. The trained networks were then evaluated on the testing HeLa cell dataset containing 100 images and testing CHO cell dataset containing

288 images, respectively. The average pixel-wise F1 scores were computed over all pixels in the two testing sets as shown in Table 2. It can be observed in the table that, on both the two datasets, E-U-nets trained with a dice+focal loss produced higher average pixel-wise F1 scores than those trained with a dice loss or a WCE loss. This comparison is now added to the revised Supplemental Information.

Loss	Dice + focal	Focal	Dice	WCE
HeLa	81.2%	81.4%	80.8%	79.4
CHO	88.0%	87.4%	87.0%	85.2%

Table 2. Average F1 scores related to E-U-nets trained with various loss functions.

We further compared E-U-nets trained with a dice+focal loss to those trained with a dice loss or a WCE loss by investigating their agreements on the dice coefficients of each class related to the predictions for each image sample in the two testing datasets. Here, let us denote $D_{\text{dice+focal}}$, D_{dice} , and D_{WCE} as the dice coefficients produced by E-U-nets trained with a dice+focal loss, a dice loss and a weighted cross entropy loss, respectively. Bland-Altman plots were employed to analyze the agreement between $D_{\text{dice+focal}}$ and D_{dice} and that between $D_{\text{dice+focal}}$ and D_{WCE} on testing dataset of HeLa and that of CHO, respectively. Here, a Bland-Altman plot of two paired dice coefficients (i.e. $D_{\text{dice+focal}}$ vs. D_{dice}) produces a scatter plot x - y , in which the y axis (vertical axis) represents the difference between the two paired dice coefficients (i.e. $D_{\text{dice+focal}} - D_{\text{dice}}$) and the x axis (horizontal axis) shows the average of the two dice coefficients (i.e. $(D_{\text{dice+focal}} + D_{\text{dice}})/2$). μ_d and σ_d represent the mean and standard deviation of the differences of the paired dice coefficients over the image samples in a specific testing dataset. The results corresponding to $D_{\text{dice+focal}}$ vs. D_{dice} and $D_{\text{dice+focal}}$ vs. D_{WCE} are reported in Figure 2 and Figure 3, respectively. In each figure, the subplots from left to right show the Bland-Altman plots related to the predictions for live, dead, and background classes, respectively. It can be observed from Figures 2-3 below that, for predicting live and dead pixels, both the $D_{\text{dice+focal}} > D_{\text{dice}}$ (or $D_{\text{dice+focal}} - D_{\text{dice}} > 0$) and $D_{\text{dice+focal}} > D_{\text{WCE}}$ (or $D_{\text{dice+focal}} - D_{\text{WCE}} > 0$) hold at the majority of the image samples in the two datasets, though for the background prediction, $D_{\text{dice+focal}}$ is comparable to D_{dice} and D_{WCE} . These results suggest that compared to a dice or WCE loss, a focal+dice loss can improve the performance of predicting

live and dead pixels for the majority of testing images from both the two datasets. Again, we added this comparison to our newly revised Supplemental Information.

Figure 2. $D_{\text{dice+focal}}$ vs. D_{dice} on testing dataset of HeLa (A) and CHO (B), where μ_d and σ_d represent the mean and standard deviation of $D_{\text{dice+focal}} - D_{\text{dice}}$.

Figure 3. $D_{dice+focal}$ vs. D_{WCE} on testing dataset of HeLa (A) and CHO (B), where μ_d and σ_d represent the mean and standard deviation of $D_{dice+focal} - D_{WCE}$.

Besides the comparison experiments, we have also moved loss function description from supplementary information into the manuscript. Please refer to the Methods in the revised manuscript.

Is the prediction of thresholded fluorescence staining better than the direct prediction of raw fluorescence image? The description of “object-based evaluation metric” is missing.

As addressed in the previous comment, the end goal of our studies is not to simply predict fluorescence stains, but to extract viability markers associated with individual cell. Since this information was usually obtained from fluorescence microscopy, we used them to generate the ground truth data, which is equivalent to manual annotation/labeling in standard deep learning applications. The output of PICS prediction is the semantic label maps, not “thresholded” fluorescence staining.

Figure 4a below shows a raw E-U-Net output image. As indicated by the yellow arrow, there exist cases where a segmented cell may have multiple semantic labels. The conventional deep learning evaluation method only focuses on assessing pixel-wise segmentation accuracy, which overlooks some biologically relevant instances (the viable state of the entire cell) [16]. And this motivates us to adopt an object-based evaluation that estimates the E-U-Net accuracy for individual cell.

In order to do this, we first used the dominant semantic label across a cellular region to denote the viable state for this cell, as shown in Figure 4b. And we compare this semantic label with the same cell in ground truth image and repeat this step across all testing images to obtain the cell-wise evaluation as shown in the article Table 2.

In revision, we now added this discussion and Figure 4 to the Supplemental information Section 3.

Figure 3. a. Output of E-U-Net on a representative testing image. The network assigns semantic labels to each pixel, and thus for some cells, more than one semantic label can be observed within the cell body. b. we use the dominant semantic label to indicate the viability state of a cell, and then the performance of training is evaluated at a cellular level, called cell-wise evaluation. Scale bars: 50 μm .

How was the training/validation/testing splitting done? Is the testing dataset sample from all “data batches” or does the testing dataset come from different “data batch” (“We repeated this experiment for four times to capture the variability among different batches.”)? Testing should be done on data from a different batch; otherwise, it can falsely increase the accuracy of the proposed method.

We thank the reviewer for the opportunity to clarify this important point further. We collected images from different batches of cells to generate the data for PICS training, validation, and testing. All image sequences were combined, where each sequence is a time-lapse recording of cells from live to dead states. Then we randomly split the sequences with a ratio of approximately 6, 1, and 1. Instead of splitting by frame, we generated training dataset by diving image sequences to ensure fair generalization.

In addition, the purpose of combining image data from different cell batches is to increase the training size. In our estimation, the E-U-Net with 75 million parameters requires approximately 2000 instances for training, and data augmentation techniques, such as random flipping and cropping, were employed in the training process. More importantly, combining data from different experiments take into account under-represented cellular activities, which makes the proposed method generalizable. In our experience, cells might present different proliferation characteristics under same preparation protocols.

In the revised manuscript, we now clarified the procedure of generating training dataset on pp. 6.

For these reasons, I do not recommend this article for publication.

We found the reviewer's criticism constructive and helpful in improving our manuscript. We hope that following our clarifications regarding novelty, our additions and revisions, the reviewer will recognize the value of our method to the field of cell biology. To our knowledge, this is the first method that allows real-time live/dead assays on unlabeled cells.

Reviewer #2 (Remarks to the Author):

Review of NCOMMS-21-12345, “Live-dead assay on unlabeled cells using phase imaging with computational specificity,” by C. Hu et al. This manuscript presents application of quantitative phase imaging to evaluating cell viability based on a deep learning approach, termed computation specificity. This work is motivated by suggestion that viability reagents decrease cell viability. Offering a non-invasive alternative could have significant benefits. This is a nice demonstration of the utility of QPI and illustrates the powerful combination with deep learning techniques.

We thank the reviewer for carefully reading our manuscript, correctly summarizing the work, providing constructive comments, and for supporting the publication of our paper. We now address the comments below.

One requirement for this type of algorithm is a large number of input data. Some discussion of how long it took to acquire these data would be a helpful inclusion. The methods section gives the exposure time but given the large number of cells examined, there must have been some variation in what time the cells were imaged vs. time after exposure. Claiming this as a high-throughput method, as done in the discussion, should be backed up with some details on imaging throughput.

We thank the reviewer for this comment. The cells were cultured in six-well plates. For each well, we randomly selected a cellular region approximately $800 \times 800 \mu\text{m}^2$ to be measured by SLIM and fluorescence microscopy (NucBlue and NucGreen). The acquisition time of each SLIM and fluorescent measurement are 50 ms and 400 ms, respectively, and the scanning across all six-wells takes roughly 4.3 minutes, where the delay is caused by mechanical translation of the motorized stage. However, we would like to point out that the acquisition of the fluorescence data is needed only for the training stage. For inference, our acquisition is up to 15 frames per second for SLIM images, while the inference takes place in parallel, i.e., by the time a SLIM image is created from the 4 individual frames, the inference is computed on the previous frame (65 ms for inference!).

In revision, we now added this clarification to the Methods section.

The illustration of loss vs number of epochs is not very illuminating. The loss appears to drop to <2% immediately and then doesn't decrease much. Perhaps a better scaling (log?) would highlight this better. Otherwise, it appears that there is not much point in training more than a bit.

Thanks for the suggestion. We now plot the log of focal loss to get a better scale in curve for the loss vs. number of epochs. Please refer to Figure 2d and Figure S4b. in the revised manuscript.

The results appear quite good but a little more examination of how and why the algorithm fails would be a good inclusion. That is Fig 3b,c only show correct classification but readers may wish to understand the limitations. It is discussed but only briefly in the results section. Some discussion on how the pixel-wise evaluation relates to the dominant semantic label would also help.

We thank the reviewer for this valuable comment. We looked back at the testing data and inspected PICS classification errors. For most of the cases, the incorrect predictions were caused by cells located at the boundary of the field of view (FOV). For instance, as shown in Figure 3, the inconsistency between the ground truth and PICS only occurs to cells where only a portion of the cell bodies were measured by SLIM. For the other cases, PICS failed when cells detached from the well plates. In this situation, the suspended cells appeared out of focus, which gives rise to inaccurate prediction. In revision, we now added these clarifications and updated Figure 3 to the manuscript (pp. 7 and 23). In addition, we included a description on object-based evaluation with a figure in the Supplemental information.

Figure 4. The cells located at the edge of the field of view are subject to erroneous classification.

The experiments with CHO cells are very similar to a recent publication from Eldridge et al (J. Biophotonics 2018) where cells were treated with staurosporine and examined with QPI. This earlier work also compared with a viability assay in the form a fret based apoptosis sensor. Some comparison with this earlier work would better place the current paper in perspective.

We thank the reviewer for pointing out this previous study. In the revised manuscript, we now cite this reference with a few other related studies. We also added a comparison of our study with the previous reported results in the Discussion Section.

The E-U-Net performance (Table 2) shows that the algorithm only correctly identifies 90% of live cells as live and nearly 10% as dead. Some discussion of this aspect, its potential impact on utility, and possible pathways to improve performance would also strengthen the paper.

We thank the reviewer for prompting us to discuss this result further. We also visually inspected the difference between the ground truth and machine learning prediction in the testing dataset. First, we saw prediction errors due to cells located at the boundary of the FOV, as explained in the previous comments. In addition, we found rare cases where live CHO cells were mistakenly labeled as dead (see Fig. 4 below for an illustration of CHO cells with staurosporine administration at $t = 0.5$ hour). In SLIM, these cells present features of abnormal cell shapes and decreased phase values, but severe membrane rupture was not observed. Previous studies suggested that these morphological features are early indicators of cell death [1, 17, 18], but it was identified as live using traditional fluorometric evaluation. In revision, we added this clarification and Fig. 6 to the Results (pp. 8) and Supplemental Information (pp. 5), respectively.

Figure 5. Cells with irregular shapes but no severe membrane rupture are subjected to erroneous classification.

The results showing that cell staining results in a decrease in area and dry mass are interesting and reinforce the need for this label-free assay. Some discussion of the increase in these parameters, which occurs for both treated and untreated, should also be included.

In this experiment, both viability reagent-labeled and unlabeled cells HeLa were placed in room temperature and low CO_2 concentration, such that this abnormal incubation condition would injure the cells and lead to necrotic cell death. This process is characterized by initial cytoplasmic and organelle swelling, followed by the loss of cell membrane integrity and release of the cellular contents into the surrounding extracellular space [19, 20]. Our tracking results suggested that, for both treated and untreated cells, the cellular area and dry mass

increased at the beginning and decreased afterwards. These results agree with physiological description [19, 20], and are consistent with previous reported experimental validations [18, 21]. However, the unlabeled and labeled cells show different rates, which indicates potential adverse effect of these chemical compounds.

In revision, we now added these clarifications to the Results section (pp. 10).

In summary, this paper presents an advance in using QPI imaging for analyzing cell features. The advance here is the use of deep learning to identify cell viability. The study is well presented and the manuscript is well written. I recommend publication after the above minor revisions.

The thank the reviewer for the encouraging assessment and valuable comments.

Reviewer #3 (Remarks to the Author):

In this article, the authors presented a label-free cell viability assessment approach, in which a deep neural network was trained to predict the segmentation maps of the live/dead cells from the quantitative phase images measured using spatial light interference microscopy (SLIM). This approach was demonstrated by imaging live adherent CHO and HeLa cells over time. By comparing the network predictions with the ground truth, a cell viability prediction accuracy of 95% was achieved. The cell nucleus area and dry mass were further evaluated based on the segmentation maps. The work is timely, very exciting, clearly organized and well presented. Furthermore, it would be of broad interest for the interdisciplinary readership of Nat Comm. Therefore, I recommend its publication after the following comments are addressed by the authors.

We truly appreciate the reviewer's insightful comments, positive assessment of our work, and support for publication. We address reviewer's comments below.

The presented neural network was trained and tested with labelled cells, therefore the reported performance of PICS method was not truly label-free. In this work, the authors added the fluorescent reagents to the cell culture before acquiring SLIM images. There are two issues that need to be considered here:

1. The viability stains can potentially change the cell morphology. This was also indicated in the article, that the labeled HeLa cells appear smaller in size compared to their label-free counterparts. Therefore, the network model, which was optimized with the images of labeled cells, perhaps may not generate entirely reliable predictions of unlabeled cells (Fig. 5d).

We thank the reviewer for giving us the opportunity to discuss this important aspect. We would like to better clarify the implication of Figure 5. In Figure 5e-f, our tracking results show that, for both treated and untreated cells, the cellular area and dry mass increased at the beginning and decreased afterwards. The increase corresponds to the cytoplasmic and

organelle swelling at the initial stage of necrotic process, and the decrease indicates membrane rupture and release of the cellular contents into the surrounding extracellular space [18-21]. The short swelling time in the reagent-treated cells suggest the toxicity of the chemical compounds would potentially accelerate the pace of cell death. We added this clarification to the revised manuscript to Results (pp. 10).

2. The viability stains might alter the cells' optical properties thus introduce additional information to the captured SLIM images. This additional information may help the network differentiate live/dead cells, but is not available for the unlabeled cells during blind testing. The reported accuracy of 95% was calibrated with images of labelled cells, which does not represent the general performance of this method on unlabeled cells.

As a suggestion, to better demonstrate the feasibility of the method, the phase images could perhaps be captured before adding the viability reagents. Then quickly perform viability labeling and capture the fluorescence images, which could serve as ground truth. Based on this, the accuracy and other metrics could be calculated. Since the cells are already plated, the alignment should not be a problem.

We agree with the reviewer that training on cell images with reagents may potentially introduce bias when optimizing the E-U-Net. However, as suggested by the reviewer, if we add viability reagents after taking QPI images, we only obtain the viability ground truth at the last time moment (all dead cells), while we lost all the viable information for the same cell before this timeframe, especially the transition between live and dead. On the other hand, we believe that the effect of the fluorescent dye itself to the optical properties of the cell at the imaging wavelength is negligible. We added 40 μ L of NucBlue® Live reagent and NucGreen® Dead reagent to 1 mL of cell growth media, where the chemical compounds were greatly diluted, and the changes to optical pathlength can be ignored. In our previous studies, we and others showed that for stained histological specimens the exogeneous dye affects the QPI signal [22, 23]. This happens because the absorption spectrum, which, through the Kramer-Kronig relation, affects the refractive index, is within the illumination spectrum of the instrument. By contrast, in fluorescence imaging, the excitation light, where the dye

absorbs strongly, is at a shorter wavelength than the imaging spectrum. Thus, for live-cells imaging, the effect of fluorescent stains onto the refractive index is negligible [24-30].

In revision, we included this point in the Discussion (pp. 11).

References:

1. Vicar, T., et al., *The Quantitative-Phase Dynamics of Apoptosis and Lytic Cell Death*. Scientific Reports, 2020. **10**(1).
2. van Meerloo, J., G.J. Kaspers, and J. Cloos, *Cell sensitivity assays: the MTT assay*. Methods Mol Biol, 2011. **731**: p. 237-45.
3. Ciapetti, G., et al., *In vitro evaluation of cell/biomaterial interaction by MTT assay*. Biomaterials, 1993. **14**(5): p. 359-64.
4. Kumar, P., A. Nagarajan, and P.D. Uchil, *Analysis of Cell Viability by the MTT Assay*. Cold Spring Harb Protoc, 2018. **2018**(6).
5. Altman, S.A., L. Randers, and G. Rao, *Comparison of trypan blue dye exclusion and fluorometric assays for mammalian cell viability determinations*. Biotechnol Prog, 1993. **9**(6): p. 671-4.
6. Riss, T.L., et al., *Cell viability assays*, in *Assay Guidance Manual [Internet]*. 2016, Eli Lilly & Company and the National Center for Advancing Translational Sciences.
7. Kandel, M.E., et al., *Phase imaging with computational specificity (PICS) for measuring dry mass changes in sub-cellular compartments*. Nat Commun, 2020. **11**(1): p. 6256.
8. Christiansen, E.M., et al., *In silico labeling: Predicting fluorescent labels in unlabeled images*. Cell, 2018. **173**(3): p. 792-803. e19.
9. Ounkomol, C., et al., *Label-free prediction of three-dimensional fluorescence images from transmitted-light microscopy*. Nature methods, 2018. **15**(11): p. 917-920.
10. Rivenson, Y., et al., *PhaseStain: the digital staining of label-free quantitative phase microscopy images using deep learning*. Light: Science & Applications, 2019. **8**(1): p. 1-11.
11. Rivenson, Y., et al., *Virtual histological staining of unlabelled tissue-autofluorescence images via deep learning*. Nature biomedical engineering, 2019. **3**(6): p. 466-477.
12. Nygate, Y.N., et al., *Holographic virtual staining of individual biological cells*. Proceedings of the National Academy of Sciences, 2020. **117**(17): p. 9223-9231.
13. Tan, M. and Q. Le. *Efficientnet: Rethinking model scaling for convolutional neural networks*. in *International Conference on Machine Learning*. 2019. PMLR.
14. Tan, M. and Q.V. Le, *Efficientnet: Rethinking model scaling for convolutional neural networks*. arXiv preprint arXiv:1905.11946, 2019.
15. Ronneberger, O., P. Fischer, and T. Brox, *U-Net: Convolutional networks for biomedical image segmentation*. arXiv 2015. arXiv preprint arXiv:1505.04597, 2019.
16. Caicedo, J.C., et al., *Evaluation of Deep Learning Strategies for Nucleus Segmentation in Fluorescence Images*. Cytometry A, 2019. **95**(9): p. 952-965.
17. Eldridge, W.J., J. Hoballah, and A. Wax, *Molecular and biophysical analysis of apoptosis using a combined quantitative phase imaging and fluorescence resonance energy transfer microscope*. Journal of Biophotonics, 2018. **11**(12): p. e201800126.
18. Pavillon, N., et al., *Early cell death detection with digital holographic microscopy*. PLoS One, 2012. **7**(1): p. e30912.
19. Duprez, L., et al., *Major cell death pathways at a glance*. Microbes and Infection, 2009. **11**(13): p. 1050-1062.
20. Krysko, D.V., et al., *Apoptosis and necrosis: Detection, discrimination and phagocytosis*. Methods, 2008. **44**(3): p. 205-221.
21. Rommel, C.E., et al., *Multimodal label-free in vitro toxicity testing with digital holographic microscopy*. Biophotonics: Photonic Solutions for Better Health Care Iv, 2014. **9129**.
22. Ban, S., et al., *Effect of tissue staining in quantitative phase imaging*. J Biophotonics, 2018. **11**(8): p. e201700402.

23. Majeed, H., et al., *Quantitative Histopathology of Stained Tissues using Color Spatial Light Interference Microscopy (cSLIM)*. Scientific Reports, 2019. **9**(1): p. 14679.
24. Kandel, M.E., et al., *Label-Free Imaging of Single Microtubule Dynamics Using Spatial Light Interference Microscopy*. ACS Nano, 2017. **11**(1): p. 647-655.
25. Kim, Y.S., et al., *Combining Three-Dimensional Quantitative Phase Imaging and Fluorescence Microscopy for the Study of Cell Pathophysiology*. Yale J Biol Med, 2018. **91**(3): p. 267-277.
26. Chowdhury, S., et al., *Spatial frequency-domain multiplexed microscopy for simultaneous, single-camera, one-shot, fluorescent, and quantitative-phase imaging*. Opt Lett, 2015. **40**(21): p. 4839-42.
27. Dubey, V., et al., *Multi-modal chip-based fluorescence and quantitative phase microscopy for studying inflammation in macrophages*. Opt Express, 2018. **26**(16): p. 19864-19876.
28. Popescu, G., *Quantitative phase imaging of cells and tissues*. McGraw-Hill biophotonics. 2011, New York: McGraw-Hill. xx, 362 p.
29. Kumar, M., et al., *Digital Holographic Multimodal Cross-Sectional Fluorescence and Quantitative Phase Imaging System*. Sci Rep, 2020. **10**(1): p. 7580.
30. Park, Y., et al., *Diffraction phase and fluorescence microscopy*. Opt Express, 2006. **14**(18): p. 8263-8.

Reviewers' Comments:

Reviewer #1:

Remarks to the Author:

Can the authors explain the increase in cell dry mass during lytic cell death captured in Fig.5 f?

According

to my knowledge and commonly accepted consensus, the cell mass increase is a result of anabolic processes connected with energy consumption. In the cells undergoing lytic cell death, the membrane

potential is disrupted and proteosynthesis doesn't run. So where is the cell dry mass coming from? During cell death, cell swelling is caused by disruption of ion homeostasis and uptake of water (increase of cell area but not dry mass).

We thank the reviewer for allowing us to clarify this point of potential confusion. In the experiment shown in Fig. 5, both viability reagent-labeled and unlabeled HeLa cells were placed in room temperature with low CO₂ concentration, such that this abnormal incubation condition would physically injure the cells and lead to cell death. Supported by extremely low NucGreen signal between $t = 0$ and $t = 1.5$ hours, we anticipate that, initially, the cells will continue to develop based on the heat and CO₂ contained in the suspension. Thus, cell death is not expected to be sudden, and our data show that the fluorescent labels speed up the live-dead transition process. According to the literature, the cell death process is characterized by initial cytoplasmic and organelle swelling, followed by the loss of cell membrane integrity and release of the cellular contents into the surrounding extracellular space [1, 2]. Figures 5a-f show that, for both treated and untreated cells, the cellular area and dry mass increased at the beginning and decreased afterwards. The increase is likely due to both the "inertial" growth mentioned above and to the cytoplasmic and organelle swelling, while the decrease indicates membrane rupture. Importantly, the unlabeled and labeled cells show different rates, which indicates potential adverse effect of these chemical compounds.

Cell swelling preceding cell lysis can hardly account for the increase in cell dry mass. During this phase of cell death, pores are formed in the cell membrane, which do not cause cell rupture but an ionic imbalance leading to water uptake by the cell. Water cannot increase the cell's dry mass. Since the fluorescence images (taken after 2h) show the beginning of cell death, a stressful environment can be assumed throughout the population studied. In such a case, it is difficult to imagine anabolic reactions associated with cell mass growth on the scale shown in Figure 5. I would rather attribute the observed phenomenon to errors in cell segmentation. However, the significance of this problem is minor and does not affect the overall significance of the paper. I hereby consider all my comments answered.

Reviewer #1 (Remarks to the Author):

Cell swelling preceding cell lysis can hardly account for the increase in cell dry mass. During this phase of cell death, pores are formed in the cell membrane, which do not cause cell rupture but an ionic imbalance leading to water uptake by the cell. Water cannot increase the cell's dry mass. Since the fluorescence images (taken after 2h) show the beginning of cell death, a stressful environment can be assumed throughout the population studied. In such a case, it is difficult to imagine anabolic reactions associated with cell mass growth on the scale shown in Figure 5. I would rather attribute the observed phenomenon to errors in cell segmentation. However, the significance of this problem is minor and does not affect the overall significance of the paper.

I hereby consider all my comments answered.

We thank the reviewer for carefully reading through this manuscript, whose comments inspired us to look deeper into the issue.

We agree with the reviewer that current PICS approach is subject to segmentation error. In this study, we focus on optimizing the PICS performance in classifying live/dead markers at the cellular level. However, as shown in Table. S1 in the Supplemental Information, at the pixel level, the average F-1 score is 73%. This suggests the trained E-U-NET can reveal the cell state changes, but its performance in capturing the nucleus shape and area is not perfect. This is largely due to the low contrast between nucleus boundary and cytoplasm in injured cells.

In revision, we now emphasis this point in the Result (pp.11) and Discussion (pp.13) Section, respectively.

Reviewer #4 (Remarks to the Author):

I agree with Reviewer 1 that the explanation by the authors of cell mass increasing at that rate is not satisfying. While I agree some degree of 'inertial' growth may be possible, the PICS predicted nuclear size increases by 60% over the first hour which seems implausible. Reviewer 1 does acknowledge that this problem is minor and doesn't affect the overall paper significance. However, a better explanation could be given by the authors to explain some limitations of the approach. In

the video (supp video 1), it is apparent that there are errors in segmentation of the cell nuclei. Indeed, the authors focus their attention away from this early time period. Instead, a more circumspect view of the limitations of PICS in this light would provide a balanced discussion.

We appreciate the reviewer's constructive comment. In the revision, we now clarify the limitations of PICS segmentation error in the Results and Discussion Section, respectively.